# Network Analysis of Neurobehavioral Symptom Patterns in an International Sample of Spanish-Speakers with a History of COVID-19 and Controls

**DOI:** 10.3390/ijerph20010183

**Published:** 2022-12-23

**Authors:** Paul B. Perrin, Daniela Ramos-Usuga, Samuel J. West, Kritzia Merced, Daniel W. Klyce, Anthony H. Lequerica, Laiene Olabarrieta-Landa, Elisabet Alzueta, Fiona C. Baker, Stella Iacovides, Mar Cortes, Juan Carlos Arango-Lasprilla

**Affiliations:** 1School of Data Science, University of Virginia, 400 Brandon Ave., Charlottesville, VA 22903, USA; 2Department of Psychology, University of Virginia, 485 McCormick Rd., Charlottesville, VA 22903, USA; 3Central Virginia Veterans Affairs Health Care System, 1201 Broad Rock Blvd, Richmond, VA 23249, USA; 4Biomedical Research Doctorate Program, University of the Basque Country, Barrio Sarriena, s/n, 48940 Leioa, Spain; 5Biocruces Bizkaia Health Research Institute, Cruces Plaza, 48903 Barakaldo, Spain; 6Department of Psychology, Virginia State University, 1 Hayden St., Petersburg, VA 23803, USA; 7Departments of Psychology and Physical Medicine and Rehabilitation, Virginia Commonwealth University, 907 Floyd Ave., Richmond, VA 23284, USA; 8Center for Traumatic Brain Injury Research, Kessler Foundation, 120 Eagle Rock Avenue, East Hanover, NJ 07936, USA; 9Health Sciences Department, Public University of Navarre (UPNA), Cataluña, s/n, 31006 Pamplona, Spain; 10Instituto de Investigación Sanitaria de Navarra (IdiSNA), 31008 Pamplona, Spain; 11Center for Health Sciences, SRI International, 333 Ravenswood Ave, Menlo Park, CA 94025, USA; 12Brain Function Research Group, School of Physiology, Faculty of Health Sciences, University of the Witwatersrand, Johannesburg 2193, South Africa; 13Department of Rehabilitation and Human Performance, Icahn School of Medicine at Mount Sinai, 1 Gustave L. Levy Pl, New York, NY 10029, USA

**Keywords:** COVID-19, SARS-CoV-2, Long-COVID, network analysis, neurobehavioral symptoms

## Abstract

(1) Background: Psychometric network analysis provides a novel statistical approach allowing researchers to model clusters of related symptoms as a dynamic system. This study applied network analysis to investigate the patterns of somatic, cognitive, and affective neurobehavioral symptoms in an international sample of Spanish-speaking individuals with a history of COVID-19 positivity and non-COVID controls; (2) methods: the sample (n = 1093) included 650 adults from 26 countries who reported having previously tested positive for COVID-19 (COVID+) through a viral and/or antigen test (average of 147 days since diagnosis). The control group (COVID−) was comprised of 443 adults from 20 countries who had completed the survey prior to the COVID-19 pandemic; (3) results: relative to the COVID− network, the COVID+ network was very well-connected, such that each neurobehavioral symptom was positively connected to the network. The organize-to-headache and dizzy-to-balance connections in the COVID+ network were stronger than in the COVID− network. The hearing, numbness, and tense symptoms were more central to the COVID+ network with the latter connected to the sleep, fatigue, and frustrated symptoms. The COVID− network was largely disjointed, with most of the somatosensory symptoms forming their own cluster with no connections to other symptom groups and fatigue not being connected to any other symptom. The cognitive and affective symptoms in the COVID− network were also largely connected to symptoms from within their own groups; (4) conclusions: These findings suggest that many of the long-term neurobehavioral symptoms of COVID-19 form a discernable network and that headaches, frustration, hearing problems, forgetfulness, and tension are the most central symptoms. Cognitive and behavioral rehabilitation strategies targeting these central symptom network features may hold promise to help fracture the lingering symptom network of COVID-19.

## 1. Introduction

Patients coined the term “Long COVID” to describe a longer and more complex course of illness than demonstrated in initial reports [1]. The medical community has generally defined Long COVID (more formally “persistent post-COVID syndrome”) as a set of symptoms that occurs after COVID-19 infection and that persists for four or more weeks [2,3]. A meta-analysis estimated that 80% of individuals with COVID-19 developed one or more symptoms after the acute phase [4]. The most common lingering COVID-19 symptoms included fatigue (58%), headache (44%), hair loss (25%), and dyspnea (24%). Other neurobehavioral symptoms included attention disorder (27%), anosmia (6%), ageusia (4%), anxiety and/or depression (4%), and memory loss (3%) [4]. Individuals during and after COVID-19 infection have reported experiencing “brain fog”—a colloquial term describing a compilation of symptoms including mental fatigue, recurrent confusion, forgetfulness, slow thinking, and disorientation [5,6]. In addition to the ongoing symptoms, more than one-third of individuals report feeling ill or in worse clinical condition than at the onset of COVID-19 [7,8,9,10,11,12].

These effects may be related directly to common acute COVID-19 symptoms [4] or to a variety of serious neurological conditions that have been associated with COVID-19 [13,14,15,16,17,18,19]. Researchers hypothesize several mechanisms through which COVID-19 affects the brain: (a) direct virus infection to astrocytes [20,21], (b) overreaction of the immune system (cytokine storm) [22], and (c) restricted blood flow in the brain causing hypoxia [23]. These mechanisms may lead to short- and long-term neurobehavioral sequelae [11,24]. Researchers argue that other multifactorial processes may mediate (directly or indirectly) the effects of the Severe Acute Respiratory Syndrome Coronavirus 2 (SARS-CoV-2) neuropathogenetic mechanisms on the central nervous system (CNS), particularly psychiatric symptoms such as anxiety and depression [25]. Hampshire and colleagues [26] found that individuals who had recovered from COVID-19 (N = 12,689) performed worse on several cognitive tests measuring reasoning, problem solving, and spatial planning than individuals who had not contracted the virus. Another study [27] of 29 individuals 12–16 weeks after hospital discharge for COVID-19 found that 59–65% of the sample presented with clinically significant cognitive impairment, the most affected being verbal learning and executive function domains. Frontera and colleagues [28] found that individuals with a history of COVID-19 reported anxiety, mental confusion, difficulty concentrating, and forgetfulness; other studies similarly have reported overlapping cognitive impairment and psychological symptoms (i.e., depression, anxiety) [29,30,31].

While these studies have documented particular types of neurobehavioral impairments in individuals with a history of COVID-19, no research has yet to investigate the constellation of lingering COVID-19 neurobehavioral symptoms as an integrated system or network. Psychometric network analysis provides a statistical approach which allows researchers to model clusters of related symptoms as a dynamic system [32,33]. It is highly likely that there is an interacting and dynamic nature of the relationships among Long COVID symptoms that can only be uncovered with novel and advanced methodological approaches like network analysis. For example, cognitive symptoms could affect mood, which disturbs sleep or engenders somatosensory disturbances that exacerbate cognitive symptoms, among other potentially mutually reinforcing feedback loops. Unfortunately, previous studies and even clinical observations are sorely lacking in this area, preventing an understanding of the overall network structure of Long COVID symptoms. Further, very little research on Long COVID has been conducted with an international scope, or specifically among Spanish speakers, limiting generalizability of findings. Accordingly, the current study used network analysis to examine the interconnections among lingering COVID-19 neurobehavioral symptoms (i.e., somatic, cognitive, and emotional) [34] in a large international sample of Spanish-speaking individuals with a history of COVID-19 versus a sample of individuals who were never infected.

## 2. Materials and Methods

### 2.1. Participants

The inclusion criteria to participate in the present study were: (a) age 18 years or older, (b) Spanish-speaking, and (c) self-report of having tested positive for COVID-19 through a viral and/or antigen test (for the COVID+ group). The sample consisted of 1,093 adults; 650 of these participants from 26 countries reported having previously tested positive for COVID-19 (COVID+) through a viral and/or antigen test (range: 1–383 days since positive test). The remainder consisted of 443 adults from 20 countries who participated prior to the COVID-19 pandemic (COVID−) (Figure 1). See Table 1 for participant demographic information.

### 2.2. Measures

Participants completed items assessing demographic information and in the COVID-19 group, days since COVID-19 diagnosis.

The Neurobehavioral Symptom Inventory (NSI) [35] is a self-report assessment of cognitive, affective, and somatic neurobehavioral symptoms. It consists of 22 items scored on a 5-point Likert scale according to severity (0 = None; 1 = Mild; 2 = Moderate; 3 = Severe; 4 = Very Severe). The NSI was administered in Spanish [36].

### 2.3. Procedure

A Qualtrics survey was distributed online through (a) professional mailing lists and collaborators’ contact networks; (b) Facebook groups and advertisements; and (c) a COVID-19 patient database from one of the collaborating centers. The study was approved by the Ethics Committee of the Public University of Navarra (PI-003/21). An informed consent document specified that participation was voluntary, data were anonymous, and there was no financial compensation for participation. The study was conducted in compliance with the Declaration of Helsinki.

### 2.4. Data Analyses

Psychometric network analysis was applied to the 22 items of the NSI. In the current network analysis, nodes represented cross-sectional variables (i.e., individual NSI symptoms) and edges represented the regularized partial correlation between any two nodes. We also estimated the strength centrality of each node in our networks. Strength centrality refers to the overall influence of a node within a network (i.e., the absolute sum of all edges connecting to a given node). We estimated our networks using the estimateNetwork function from the bootnet package for R version 4.1.1 [37,38]. We implemented the EBICglasso routine which attempts to finds a relatively sparse network that best fits the data by reducing the smallest edges in the network to zero and examining the model fit until maximal fit is achieved [37,38]. Given the Likert-response nature of our data, we applied polychoric correlations in our network estimation. Network stability and accuracy were examined by two 1000-sample bootstraps (i.e., case-dropping and non-parametric) using the bootnet function from the bootnet package [37,38]. The survey software required that all questions be answered for participants to proceed, so no data were missing. The average network layout was computed using the averageLayout function from the qgraph package prior to visualization to aid in comparison [39,40]. Finally, we compared our networks (COVID+ versus COVID−) statistically using a 1,000-iteration permutation test from the NetworkComparisonTest package [41].

## 3. Results

Descriptive statistics are presented in Appendix A. No variables in the current study demonstrated substantial skew, excepting the Appetite node among the COVID− participants. Applying the nonparanormal transformation to this variable did not improve the skew, and thus the original data were retained [42]. We screened both datasets separately for univariate outliers (i.e., beyond +/− 3SD from the mean), which were Winsorized prior to network estimation. 

### 3.1. Network Structure

Correlation stability coefficients were ideal for edge weights (*CS* = 0.75) and strength centrality estimates (*CS* = 0.67) in the COVID+ network. Similar stability was found in the COVID− network for edges (*CS* = 0.44), but strength centrality was relatively lower (*CS* = 0.36), though still acceptable. Both networks demonstrated accuracy in relation to the estimated edges, as only a single edge (Clumsy—Organize from the COVID+ network) was absent in more than 50% of the nonparametric bootstrap samples. Full edge weight estimates and related bootstrapped values are presented in Appendix A. The COVID+ network emerged as a relatively sparse, yet well-connected, network (33/231 possible edges) such that each symptom assessed by the NSI was connected to the network. The COVID+ network also evinced many connections in the three symptom groups, and all edges in the COVID+ network were positive in direction (Figure 2). Conversely, the COVID− network yielded relatively fewer edges (27/231 possible edges), and the Fatigue node was not connected to any other node in the network. The COVID− network was largely disjointed, with most of the Somatosensory symptoms forming their own cluster with no edges to the other symptom groups. The Cognitive and Affective items in the COVID− network were also largely connected to symptoms from within their own groups except for four edges: the Appetite—Organize edge and the edges shared among the Head node and the Organize, Forget, and Decide nodes (Figure 3).

### 3.2. Global Network Comparisons

Our network comparison test revealed that the structure of the COVID+ network differed significantly from the COVID− network, M = 0.30, *p* = 0.010. We also compared the global strength of the networks, which failed to yield a significant difference, S = 1.34, *p* = 0.074.

### 3.3. Edge Comparisons

Examination of the specific edge comparisons resultant from the network comparison test revealed several notable differences. The network comparison test identified nine significantly different edges between the networks. As such, we only discuss the two significant differences of edges that appeared in both networks here (see Appendix A for all edge comparisons). Of these, the largest difference between networks was the Organize—Head edge. In the COVID+ network, this edge had a weight of 0.41, but was 0.18 in the COVID− network, less than half the associative strength, *p* < 0.001. The Dizzy—Balance edge was also significantly different across networks. In the COVID+ network this edge yielded a weight of 0.47, compared to 0.32 in the COVID− network, *p* = 0.023.

### 3.4. Centrality Comparisons

To quantify the similarity of centrality implied by the global strength invariance test, we conducted a Spearman’s rank-order correlation analysis using the strength estimates from each network. This analysis indicated that less than half of the total variance in centrality overlapped between networks, ρ = 0.69, *p* < 0.001. We thus explored the individual strength centrality comparisons across networks despite our marginal global test. See Table 2 for all strength centrality estimates and comparison *p*-values. The network comparison test revealed four significant differences. First, the Tense node yielded significantly greater strength centrality in the COVID+ network, *p* = 0.029. In the COVID− network, this node was connected only to the Frustrated node, where in the COVID+ network it was connected to the Sleep and Fatigue nodes in addition to the Frustrated node. As such, this suggests that feelings of tension among the COVID+ network played a crucial role in connecting the Affective symptoms measured by the NSI. Next, the Hearing node had significantly greater strength centrality in the COVID+ network, *p* = 0.034. As observed with the Tense node, this was due to more dense connections with other symptoms within its own group (i.e., Somatosensory symptoms; Vision, Nausea, Numb, and Light). Third, the Numb node was significantly more central in the COVID+ network, *p* = 0.038. This difference appeared to be due largely to the Numb—Forget edge that emerged in the COVID+ network, but not COVID− network. Finally, the Fatigue node was significantly more central to the COVID+ network, *p* < 0.001, because it shared no connections with the COVID− network and thus had a centrality estimate of zero.

## 4. Discussion

This study applied psychometric network analysis to investigate the patterns of somatic, cognitive, and affective neurobehavioral symptoms in an international sample of Spanish-speaking individuals with a history of COVID-19 and non-COVID controls. A large sample of 1,093 individuals (650 COVID+ and 443 COVID−) completed an online self-report survey assessing their neurobehavioral symptoms. Relative to the COVID− network, the COVID+ network was very well-connected such that each neurobehavioral symptom was positively connected to the network. The COVID− network was largely disjointed, with most of the somatosensory symptoms forming their own cluster with no connections to other symptom groups and fatigue not being connected to any symptom. The cognitive and affective symptoms in the COVID− network were also largely connected to symptoms from within their own groups.

As an application of dynamic systems theory, psychometric network analysis can be a compelling approach to understand the associations among constellations of symptoms that make up clinical syndromes. The most important difference the analysis uncovered between the two networks was the coherence and well-connectedness of the COVID+ network compared to the disjointed COVID− network. This difference is likely due to the fact that a COVID− subsample by definition would not have patterns of lingering COVID-19 neurobehavioral symptoms. As a result, the subsample’s symptoms, when present, would be due to other potential health conditions or disabilities and only interrelate if the symptoms were extremely similar. In contrast, the COVID+ network showed tremendous cross-talk across symptom types and greater connectivity overall, indicating that any given symptom has greater associations with other symptoms because of a history of COVID-19. This analysis—for the first time in the literature—identified an interrelated network of lingering COVID-19 symptoms above and beyond those expected in a COVID− sample simply because specific neurobehavioral symptoms are similar. When compared to the COVID− network, the coherence of the COVID+ network suggested the utility of conceptualizing long-term neurobehavioral symptoms as a post-COVID syndrome.

In terms of specific relations among symptoms, the organize-to-headache bridge in the COVID+ network was stronger than in the COVID− network. Individuals with a history of COVID-19 uniquely showed a very strong connection between headaches and organization; while the COVID− sample showed the same edge, it was much weaker, perhaps reflecting the difference between a headache that is annoying/distracting among controls versus one that is truly debilitating among individuals with a history of COVID-19. Because the data were cross-sectional, it is unknown whether headaches cause organizational problems after COVID-19, or perhaps whether some other unknown variable (e.g., brain lesions) causes both. Similarly, the dizzy-to-balance bridge was stronger in the COVID+ network, consistent with emerging literature demonstrating vestibular symptoms being a common component of COVID-19 [43]. In the COVID+ sample, the concentrate symptom was connected to five other symptoms (organize, appetite, fatigue, vision, and clumsy), but in the COVID− sample, it was only connected to the decide symptom. As a result, in Long COVID, concentration problems are likely extremely interrelated to affective, cognitive, and somatosensory symptoms, presenting multiple clinical targets that if improved, might increase concentration ability. Further, in the COVID+ network, the fatigue symptom was connected to taste (and smell), appetite, decide, and tense symptoms, though in the COVID− sample it was unconnected to any other symptom. In Long COVID, fatigue may be stress-related (tense) and due to nutrition (appetite and taste/smell), as well as connected to the ability to make decisions.

### 4.1. Connections with Previous Research and Implications

Lingering, post-COVID concerns commonly include depressed mood, insomnia, anxiety, irritability, fatigue, and “brain fog” or memory impairment [31,44]; indeed, a subjective sense of persistent cognitive impairment is a central feature [45,46]. The current findings suggest the high degree to which these symptoms co-occur after COVID-19. Among such patients who participated in comprehensive neuropsychological assessment, Krishnan and colleagues [46] found a general pattern of mild impairment on measures of processing speed, attention, and executive functioning. Of note, 70% of these authors’ participants also had a history of mood disorder prior to infection, and approximately 35% of their sample endorsed moderate-to-severe mood symptoms at the time of assessment. The current study found hearing, numbness, and tense symptoms were more central to the COVID+ network with the latter connected to sleep, fatigue, and frustrated symptoms. These current findings in light of those from Krishnan and colleagues implicate an affective (e.g., mood, tension, frustration) component associated with post-COVID syndrome and suggest a potential role for psychotherapeutic and rehabilitation strategies to disrupt this network of affective distress [46]. Given the potential benefit of conceptualizing post-COVID neurobehavioral symptoms as a syndrome, the current findings also suggest that the nodes with greater strength centrality estimates could be key targets for intervention—i.e., headaches, poor frustration tolerance, hearing difficulty, forgetfulness, and feeling anxious or tense.

The strength of the association among the central symptoms and other items on the NSI provide a framework for conceptualizing what may be clusters of symptoms experienced by individuals with a history of COVID-19. As our understanding of persistent COVID-19 symptoms evolves, these constellations of symptoms can be further investigated to examine premorbid factors that contribute to greater vulnerability to ongoing complications so that affected individuals can be triaged for specialized treatment in a patient-centered model of care. There are a variety of biological, psychological, and environmental factors that likely contribute to persistent COVID-19 neurobehavioral symptoms (e.g., sensitivity of multiple organ systems to COVID-19 that could affect cognitive health; pre-infection psychological and social vulnerabilities; and pandemic-related stress and isolation) [47]. Rehabilitation clinicians would likely benefit from adopting a comprehensive approach targeting as many of these potential influences on neurobehavioral symptoms as possible. Cognitive and behavioral rehabilitation strategies targeting these central symptom network features may hold promise to help those experiencing unresolved post-COVID symptomatology.

### 4.2. Study limitations and Future Directions

Despite the importance of these findings, the current study had several limitations, and as a result, directions for future research. First, while the NSI has been used primarily among individuals with concussion, the items contain a wide variety of symptoms across somatic, affective, and cognitive domains, and the rating structure includes the degree to which items interfere with functioning, all of which may be useful to clinicians treating a wide variety of neurological conditions. However, there may be additional symptoms specific to COVID-19 not tapped by the NSI that future similar network analysis research should incorporate. Second, the current Spanish-speaking sample was from an extremely wide array of over 20 countries, and as a result, the findings have a high degree of generalizability, far more than in traditional studies. Nonetheless, the internet-based data collection presumed that participants had access to the internet and were fluent in Spanish. As a result, the findings may have limited generalizability to participants without internet access or who speak languages other than Spanish. Third, additional demographic information, such as current work status, could have shed additional light on participants’ symptom patterns. For example, participants in the COVID+ group who were working may have encountered cognitive challenges as they adjusted back to the demands of full-time work after their acute COVID period. Future research should assess and consider the influence of other important demographics on Long COVID symptom networks. Finally, the data were cross-sectional and correlational, and as a result, the causal influence of symptoms on each other cannot be ascertained. Future longitudinal research can better tease apart the relative causal influence of COVID-19 symptoms on each other.

## 5. Conclusions

Notwithstanding these limitations, this study was the first to identify a coherent network of post-COVID-19 neurobehavioral symptoms and to compare the network to that of a group of individuals without a history of COVID-19. These findings suggest that many of the long-term neurobehavioral symptoms of COVID-19 form a discernable network and that headaches, frustration, hearing problems, forgetfulness, and tension are the most central symptoms. Cognitive and behavioral rehabilitation strategies targeting these central symptom network features may hold promise to help fracture the lingering symptom network of COVID-19.

## Figures and Tables

**Figure 1 ijerph-20-00183-f001:**
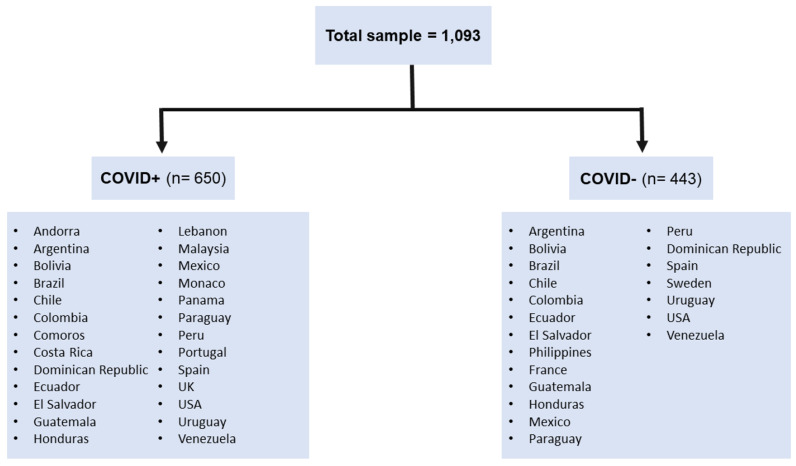
Countries included in the sample.

**Figure 2 ijerph-20-00183-f002:**
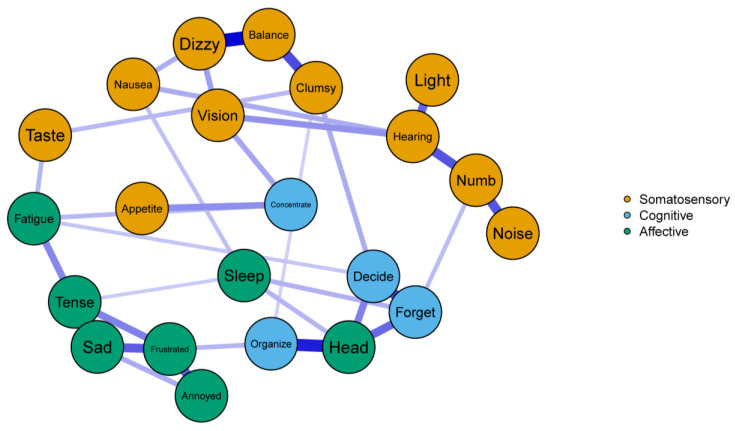
The COVID+ network. Solid edges indicate positive associations with width and depth of color indicating the strength of associations.

**Figure 3 ijerph-20-00183-f003:**
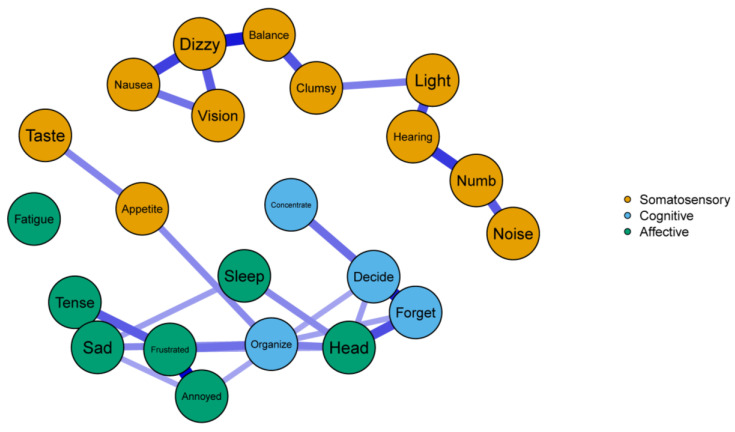
The COVID− network. Solid edges indicate positive associations with width and depth of color indicating the strength of associations.

**Table 1 ijerph-20-00183-t001:** Sociodemographic characteristics of the study cohort (N = 1093).

Variable	COVID+		COVID−	
Age (years), M ^1^, SD ^2^	43.5	11.9	33.9	13.2
Gender, N, %				
Man	130	20.0	140	31.6
Woman	519	79.8	303	68.4
Non-binary, transgender, or other	1	0.2	0	0
Highest level of education completed, N, %				
Some school education	12	1.8	54	12.4
Graduated high school	31	4.9	40	9.1
Some college/technical degree	145	22.3	69	15.7
Completed undergraduate education	201	30.9	47	10.9
Postgraduate education (some or completed)	261	40.1	230	51.9
Days from COVID-19 diagnosis, M, SD	147.6	98.9	-	-

^1^ Mean; ^2^ Standard Deviation.

**Table 2 ijerph-20-00183-t002:** Strength Centrality Estimates from Both Networks.

Node	Symptom Group	COVID+	COVID−
Annoyed	Affective	0.46	0.61
Appetite	Somatosensory	0.21	0.34
Balance	Somatosensory	0.80	0.56
Clumsy	Somatosensory	0.70	0.42
Concentrate	Cognitive	0.51	0.21
Decide	Cognitive	0.83	0.59
Dizzy	Somatosensory	0.79	0.82
Fatigue	Affective	0.60	0.00
Forget	Cognitive	0.88	0.91
Frustrated	Affective	0.97	0.92
Head	Affective	1.06	0.87
Hearing	Somatosensory	0.95	0.53
Light	Somatosensory	0.28	0.43
Nausea	Somatosensory	0.42	0.46
Noise	Somatosensory	0.31	0.23
Numb	Somatosensory	0.75	0.51
Organize	Cognitive	0.64	0.85
Sad	Affective	0.78	0.76
Sleep	Affective	0.49	0.30
Taste	Somatosensory	0.27	0.17
Tense	Affective	0.88	0.55
Vision	Somatosensory	0.54	0.42

## Data Availability

Not applicable.

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
