# Peer review of "Network Analysis of Neurobehavioral Symptom Patterns in an International Sample of Spanish-Speakers with a History of COVID-19 and Controls"

_ijerph, 2022, doi:10.3390/ijerph20010183_

Round 1
Reviewer 1 Report
Dear Authors,
Congratulations on this study in which you used the original analysis method. My suggestions and concerns about your work are in the comment boxes in the pdf file.
regards

Reviewer 2 Report
The manuscript reports the findings of the comparison of the structural relationship of various cognitive, affective, and somatic-neurobehavioral symptoms between individuals with COVID-19 and without, using a network analytic approach. The Introduction is informative and the analysis is well-planned and –executed. The findings shed new light on understanding the physical and psychological consequences of COVID-19 as a dynamic system of network. I have several minor comments for the authors to consider, as follows:
Introduction:
- There is a lack of an argument for the (potentially) interacting and dynamic nature of the relationship between these symptoms, which is the main argument for the use of network analysis. The authors may further elaborate on this (e.g. cognitive symptoms affect mood, which disturbs sleep and brings somato-sensoary disturbances and further cognitive symptoms), based on previous studies and/or clinical observations. Similarly, these dynamics between symptoms could be further elaborated in the Discussion.
Methods:
- It is mentioned that “650 of these participants from 26 countries reported having previously tested positive for COVID-19 (COVID+) through a viral and/or 104 antigen test” (lines 103-105, p. 3). A brief breakdown of the countries would help readers appreciate the diversity of nationality in this sample.
- The NSI is rated on a 5-point Likert scale. It would be informative to mention how the correlations, for example, polychoric or spearman, are modelled in the network analysis, which increases the transparency of the analysis, as well as replicability.
- The proportion of the missing values (line 140, p. 4) could be mentioned for readers’ reference.
Results:
- I recommend putting “Correlation stability coefficients were ideal… (Tables S3 – S4)” (lines 151-158, p. 4) in the respective sections 3.1 -3.3, so these findings could be better integrated with the corresponding result sections for clarity and flow.
- For Supplementary Table S2, the M column seems to refer to the number of available responses instead of means. Please double-check the name of the columns.
Discussion:
- Line 303 (p. 8) mentions that other COVID-19-specific symptoms, such as sense of smell, should be assessed. Just wondering if there is an item on “Change in taste and/or smell” in the NSI. Maybe I am wrong.
Round 2
Reviewer 1 Report
Dear Authors,
It is seen that the authors made the revisions carefully. I think the article can be published as it is.
Best regards